# Association of poverty-income ratio with cardiovascular disease and mortality in cancer survivors in the United States

Vidhushei Yogeswaran[1‡], Youngdeok Kim[2‡*], R. Lee Franco[2], Alexander R. Lucas[3], Arnethea L. Sutton[2], Jessica G. LaRose[3], Jonathan Kenyon[2], Ralph B. D'Agostino, Jr.[4], Vanessa B. Sheppard[3,5], Kerryn Reding[6], W. Gregory Hundley[7], Richard K. Cheng[1]

1 Division of Cardiology, University of Washington Medical Center, Seattle, WA, United States of America, 2 Department of Kinesiology and Health Sciences, Virginia Commonwealth University, Richmond, VA, United States of America, 3 Department of Health Behavior and Policy, School of Medicine, Virginia Commonwealth University, Richmond, VA, United States of America, 4 Department of Biostatistics and Data Science, Wake Forest University School of Medicine, Winston-Salem, NC, United States of America, 5 Massey Cancer Center, Virginia Commonwealth University, Richmond, VA, United States of America, 6 Department of Biobehavioral Nursing and Health Informatics, University of Washington School of Nursing, Seattle, Washington, United States of America, 7 Pauley Heart Center, Virginia Commonwealth University, Richmond, VA, United States of America

‡ VY and YK are co-first authors
* kimy13@vcu.edu

**Data Availability Statement:** The NHANES data are publicly available from the CDC website (https://www.cdc.gov/nchs/nhanes/index.htm).

## Abstract

### Background

Lower income is associated with high incident cardiovascular disease (CVD) and mortality. CVD is an important cause of morbidity and mortality in cancer survivors. However, there is limited research on the association between income, CVD, and mortality in this population.

### Methods

This study utilized nationally representative data from the National Health and Nutrition Examination Survey (NHANES), a cross-sectional survey evaluating the health and nutritional status of the US population. Our study included NHANES participants aged ≥20 years from 2003–2014, who self-reported a history of cancer. We evaluated the association between income level, prevalence of CVD, and all-cause mortality. All-cause mortality data was obtained through public use mortality files. Income level was assessed by poverty-income ratio (PIR) that was calculated by dividing family (or individual) income by poverty guideline. We used multivariable-adjusted Cox proportional hazard models through a backward elimination method to evaluate associations between PIR, CVD, and all-cause mortality in cancer survivors.

### Results

This cohort included 2,464 cancer survivors with a mean age of 62 (42% male) years. Compared with individuals with a higher PIR tertiles, those in the lowest PIR tertile had a higher rate of pre-existing CVD and post-acquired CVD. In participants with post-acquired CVD, the lowest PIR tertile had over two-fold increased risk mortality (Hazard Ratio (HR) = 2.17;

**Funding:** The author(s) received no specific funding for this work.

**Competing interests:** The authors have declared that no competing interests exist.

95% CI: 1.27–3.71) when compared to the highest PIR tertile. Additionally, we found that PIR was as strong a predictor of mortality in cancer survivors as CVD. In patients with no CVD, the lowest PIR tertile continued to have almost a two-fold increased risk of mortality (HR = 1.72; 95% CI: 1.69–4.35) when compared to a reference of the highest PIR tertile.

## Conclusions

In this large national study of cancer survivors, low PIR is associated with a higher prevalence of CVD. Low PIR is also associated with an increased risk of mortality in cancer survivors, showing a comparable impact to that of pre-existing and post-acquired CVD. Urgent public health resources are needed to further study and improve screening and access to care in this high-risk population.

## Introduction

Cardiovascular disease (CVD) and cancer are two of the leading causes of death in the United States (US) [1]. Both CVD and cancer exhibit an inverse association with income inequality [2, 3]. The field of cardio-oncology has emerged in response to cardiovascular complications associated with cancer therapy [4, 5], linking the underlying pathophysiology, cardiometabolic comorbidities, risk factors, and disparities observed in CVD and cancer [1, 3, 6].

Income inequality has risen significantly in recent decades, widening healthcare disparities. Today, life expectancy is 12–14 years longer for the highest income groups, compared to the lowest, a stark contrast to a 5 year-difference just a few decades ago [7]. Researchers have demonstrated that lower income is associated with an increased risk of poor health, CVD, and mortality [3, 8–10]. This widening income gap has also exacerbated CVD disparities [11] and influenced cancer risk, with higher-income individuals and countries worldwide reporting lower cancer incidence [12].

Cancer survivors exhibit a higher prevalence of cardiovascular risk factors, including obesity, hypertension, diabetes, and physical inactivity, compared to the general population [13, 14]. As a result, CVD becomes a significant cause of morbidity and mortality in cancer survivors. Several large studies support that cancer survivors have an increased risk of CVD compared to the general population [15]. Additionally, certain cancer therapies are linked to long-term adverse cardiovascular effects such as coronary artery disease, heart failure, and vascular disease [16, 17]. However, less is known about the impact of income on CVD risk and mortality in cancer survivors [18, 19].

To address this knowledge gap and evaluate how income disparities impact cardio-oncology outcomes, we performed an analysis using a nationally representative data, the National Health and Nutrition Examination Survey (NHANES), to provide insight into the relationship between income inequality, CVD, and all-cause mortality in cancer survivors. Our study aimed to 1) assess the prevalence of CVD and income equality among cancer survivors and investigate their associations, and 2) explore the role of income inequality as a factor associated between CVD status and all-cause mortality in cancer survivors.

## Methods

### Study population

We analyzed publicly available National Health and Nutrition Examination Survey (NHANES) data conducted by the National Center for Health Statistics (NCHS). The

NHANES is a cross-sectional survey evaluating the health and nutritional status of the US population in 2-year cycles based on nationally representative samples selected by a complex sampling design [20]. The NHANES examines various health-related outcomes through household interviews, physical examinations, and laboratory tests. The survey protocol of the NHANES was approved by the Research Ethics Review Board at the NCHS. Our analysis included all adult participants in NHANES with publicly available data and a history of cancer. For this study, history of cancer (aged ≥18 years) was identified based on self-reported medical history data. We considered cancer survivors to be those individuals who had been previously diagnosed with cancer and answered "yes" to the question, "Has a doctor ever told you that you had cancer or malignancy?" Cancer sub-types were assessed through response to the question "What kind of cancer?", and respondents were able to enter up to 3 kinds. Additionally, data on age at cancer diagnosis was collected through a separate survey question [21, 22].

Cancer-type was then grouped into the following categories: Obesity-related cancer, Tobacco-related cancer, and individual sub-types. Obesity-related was defined by the International Agency for Research on Cancer (IARC) to include esophageal, colorectal, gallbladder, pancreas, breast, uterine, ovarian, kidney, or thyroid cancer [23] and Tobacco-related IARC includes nasal cavity, paranasal sinus, nasopharynx, hypopharynx, oropharynx, larynx, esophageal, bladder, cervix, lung, stomach, liver, and myeloid leukemia [24], Individual cancer sub-types were separated into 1) Breast, 2) Lung, 3) Colon, 4) Prostate, 5) Melanoma, 6) Blood (Leukemia and Lymphoma), and 7) All other types of cancer. If multiple cancers are reported, the first diagnosed cancer type was considered as the primary cancer and used for categorization.

We combined the six NHANES cycles (2003–2004 through 2013–2014) to increase the sample size of cancer survivors. Of participants who provided valid response in cancer status (n = 3,106), participants with missing or invalid data on study covariates were excluded resulting in a final analytic sample of 2,464 participants with a history of cancer (see Fig 1 for a flow chart of sample selection). The excluded sample (n = 642) was not significantly different from the final analytic sample in terms of age, gender, marital status, health insurance, BMI, cancer types, or CVD status. However, the sample did exhibit a significantly lower education level, a lower proportion of non-Hispanic White individuals, and a higher level of MVPA.

To assess all-cause mortality, our sample population was linked with public use linked mortality files associated with NHANES. Mortality data included the mortality status of the

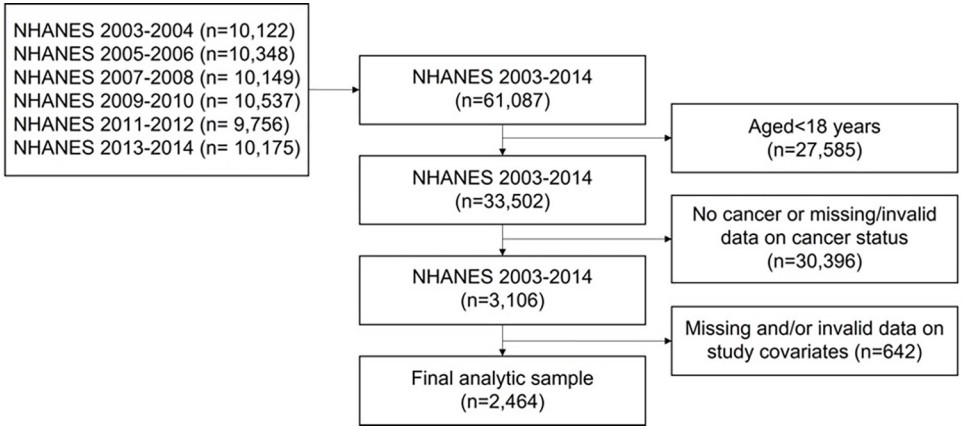

**Fig 1. Study population for NHANES to assess the impact of income on cardiovascular risk and mortality amongst cancer survivors.**

NHANES participants up to December 31, 2015, with follow-up time calculated from the date of interview to either the date of death or the end of the mortality period.

## Income measures

During a household interview, NHANES participants were asked to report total family income in the previous calendar year. This total family income is then divided by the Department of Health and Human Services (HHS) poverty guidelines threshold, specific to family size, calendar year, and state. The PIR index, developed by NHANES, accounts for regional cost of living and annual inflation. PIR has been extensively validated as a measure of income and has been documented in various studies [11, 25].

The PIR values ranged from 0 to 5 ($\geq$ 5 due to disclosure concerns). A PIR value of 0 corresponds to no income, 1 indicates a family income at 100% of the federal poverty level, and 5 corresponds to $\geq$5 times the federal poverty level. For this study, we categorized the participants into three groups based on the tertiles (Lowest tertile: PIR<1.72; Middle tertile: PIR 1.72 thru <3.71; Highest tertile: PIR$\geq$3.71).

## Measurement of covariates

Sociodemographic variables (age (years), sex (i.e., female vs. male), race/ethnicity (i.e., non-Hispanic White, non-Hispanic Black, Mexican, and others), education (i.e., <high school, high school diploma or equivalent, and college or above), marital status (i.e., married or living with a partner vs. single), and health insurance (i.e., covered vs. not covered), and health behavior variables were measured. Health behaviors included smoking status (currently smoking vs. non-smoking), alcohol consumption (i.e., <12 alcohol drinks/year vs. $\geq$12 alcohol drinks/year), body mass index (i.e., <25 kg/m$^2$, 25 thru <30 kg/m$^2$, and $\geq$30 kg/m$^2$), walking difficulty score, healthy eating index (HEI-2015 score) [26], and physical activity levels. HEI-2015 scores ranged from 0–100, with higher scores reflecting better diet quality. Walking difficulty score was calculated by averaging the responses to the three mobility limitation questions (4-point Likert scale), including 1) walking for a quarter mile difficulty; 2) walking up ten steps difficulty; and 3) walking between rooms on same floor difficulty, with greater values indicating greater walking difficulties.

Physical activity level was characterized by self-reported weekly time spent in moderate-to-vigorous physical activity (MVPA) assessed using the Global Physical Activity Questionnaire. Individuals were categorized into two groups, sufficient-MVPA ($\geq$150 minutes/week) and insufficient-MVPA (<150 minutes/week).

## Measurement of outcomes

The primary outcome was defined as self-reported CVD. All NHANES participants were asked to report their health conditions, including history of CVD, which encompasses coronary artery disease (i.e., coronary heart disease and heart attack), heart failure, and stroke diagnosed by a doctor or health professional. We categorized participants CVD into three groups: No CVD, pre-existing CVD (i.e., CVD diagnosed before cancer diagnosis), and post-acquired CVD (i.e., CVD diagnosed after cancer diagnosis). For the post-acquired CVD group, we calculated an average time of CVD diagnosis since they were first diagnosed with cancer. Individual breakdowns for CVD (coronary artery disease, heart failure, and stroke) were reported. Individuals with more than two types of CVD were reported as 2+ CVD.

The secondary outcome was defined as all-cause mortality. Mortality data was obtained using public use linked mortality files associated with the NHANES. Mortality data included

the mortality status of the NHANES participants through December 31$^{st}$, 2015, and follow-up time from the date of interview to the death or the end of the mortality period.

## Statistical analysis

Descriptive characteristics of study variables by PIR tertile were calculated using mean or percentage along with 95% confidence intervals for continuous and categorical variables, respectively. The prevalence of CVD among cancer survivors was estimated by PIR tertile. We reported the prevalence of cancer survivors with 2 or more CVDs (CAD, heart failure, and stroke) and individual CVD diagnosis (across the time when CVD was diagnosed (i.e., pre-existing before cancer diagnosis or post-acquired CVD after cancer diagnosis) as determined by self-report. A general linear model followed by pairwise comparisons was used for between-group comparison of continuous variables by PIR tertile. Rao-Scott x$^2$ test of independence was used for testing between-group differences in proportions for categorical variables. A multivariate Cox proportional hazards regression model was constructed, predicting the risk of all-cause mortality based on PIR and CVD status among cancer survivors. We first fit a model including both PIR and CVD status as primary independent variables predicting the risk of all-cause mortality. A linear trend of the risk across PIR tertiles was tested using orthogonal polynomial contrasting, and the interaction effects between PIR and CVD status were examined. The risk of all-cause mortality was presented using hazard ratios (HR) relative to the reference category (lowest tertile for PIR; and no-CVD for CVD status). The second model included the combined variable of PIR and CVD consisting of 9 combined groups (three PIR tertiles by three CVD status) to test their joint association with mortality risk. The risk of all-cause mortality was estimated relative to the highest PIR tertile/no-CVD group as the primary reference. Additionally, HRs relative to alternative groups were reported as a supplement. For Cox proportional hazards regression, the models were adjusted for study covariates retained through backward elimination. All study variables reported in Table 1 were entered into the model, and a variable with the largest *P*-value (≥.20) was sequentially excluded until reaching the final model, consisting of study covariates with a *P* < .20. The proportional hazard assumption was evaluated for all study variables included in model using Schönfeld residual and by adding an interaction term with the time in the model. All analyses accounted for complex, multistage, and probability sampling scheme of the NHANES (e.g., oversampling of minority sub-groups, survey non-response, and post-stratification adjustment) using the SURVEY procedures in SAS v9.4 (SAS Institute, Cary, NC). A 12-year sampling weight was calculated based on published guidelines [27], by dividing the 2-year sampling weight from each cycle by the number of NHANES cycles combined to estimate the population level parameters in the US. P≤.05 was considered statistically significant.

## Results

The final analytic sample included 2,464 cancer survivors, which when weighted, represented 12,583,919 cancer survivors, equivalent to 7.91% of US adults ≥18 years in age. The mean age was 61 years, with 42% male sex. On average, participants were first diagnosed with cancer 11 years prior to the study. Table 1 summarizes the baseline sociodemographic, health behaviors, and clinical characteristics by PIR tertile. There were 821 (weighted 1.78%) participants in the lowest PIR tertile, 815 (weighted 2.48%) in the middle tertile, and 828 (weighted 3.65%) in the highest PIR tertile.

Participants in the lowest PIR tertile, compared with those in the middle and highest, were less likely to be male, have health insurance, have a college education, or be married. Additionally, they were more likely to be smokers, be obese, have difficulty walking, and were less likely

**Table 1. Descriptive characteristics of the cancer survivors (aged≥18 years) by poverty-income-ratio (NHANES 2003–2014).**

| | Total | Poverty-income ratio[a] | | | P-value[b] |
|---|---|---|---|---|---|
| | | Lowest tertile | Middle tertile | Highest tertile | |
| Unweighted sample size (n) | 2,464 | 821 | 815 | 828 | |
| Weighted population size (N) | 12,583,919 | 2,836,361 | 3,943,044 | 5,804,515 | |
| Weighted population prevalence (%)[c] | 7.91 (7.48, 8.33) | 1.78 (1.57, 1.99) | 2.48 (2.26, 2.69) | 3.65 (3.33, 3.96) | |
| Average follow-up years | 5.94 (5.81, 6.07) | 5.67 (5.46, 5.88) | 5.96 (5.73, 6.19) | 6.18 (5.96, 6.40) | < .001 |
| Unweighted deaths (n) | 527 | 212 | 207 | 108 | |
| Age (years) | 61.85 (61.05, 62.64) | 60.92 (59.39, 62.44) | 64.79 (63.37, 66.21) | 60.3 (59.27, 61.33) | < .001 |
| Years since first diagnosed with cancer | 11.34 (10.72, 11.97) | 11.32 (10.50, 12.15) | 11.81 (10.90, 12.73) | 11.03 (10.04, 12.03) | < .001 |
| Sex (%) | | | | | < .001 |
| Male | 42.44 (40.3, 44.58) | 28.38 (24.61, 32.16) | 43.82 (40.58, 47.06) | 48.37 (44.67, 52.08) | |
| Race/ethnicity (%) | | | | | < .001 |
| Non-Hispanic White | 88.3 (86.64, 89.97) | 79.21 (75.58, 82.83) | 87.60 (84.73, 90.46) | 93.23 (91.66, 94.79) | |
| Non-Hispanic Black | 5.28 (4.24, 6.32) | 9.63 (7.14, 12.12) | 6.39 (4.70, 8.09) | 2.40 (1.70, 3.09) | |
| Mexican | 2.02 (1.31, 2.73) | 4.89 (2.76, 7.02) | 1.85 (0.95, 2.75) | 0.72 (0.31, 1.14) | |
| Others | 4.4 (3.43, 5.38) | 6.27 (4.20, 8.35) | 4.16 (2.41, 5.90) | 3.65 (2.32, 4.99) | |
| Types of Cancer (%)[d] | | | | | < .001 |
| Obesity-Related IARC | 9.59 (8.05, 11.14) | 11.47 (8.62, 14.33) | 8.81 (6.56, 11.06) | 9.21 (6.75, 11.67) | |
| Tobacco-Related IARC | 12.04 (10.27, 13.81) | 18.86 (15.34, 22.39) | 12.50 (9.69, 15.30) | 8.40 (5.72, 11.08) | |
| Breast | 15.07 (13.38, 16.75) | 18.37 (15.48, 21.26) | 14.59 (11.3, 17.88) | 13.77 (11.01, 16.54) | |
| Lung | 1.92 (1.27, 2.58) | 1.15 (0.17, 2.13) | 3.26 (1.62, 4.91) | 1.39 (0.52, 2.26) | |
| Colon | 4.95 (3.91, 5.99) | 7.12 (5.03, 9.22) | 5.53 (3.81, 7.26) | 3.49 (2.13, 4.85) | |
| Prostate | 8.56 (7.24, 9.88) | 6.57 (5.08, 8.06) | 10.04 (8.10, 11.99) | 8.52 (6.40, 10.63) | |
| Melanoma | 7.01 (5.53, 8.49) | 5.71 (3.93, 7.49) | 6.96 (4.42, 9.50) | 7.68 (5.24, 10.13) | |
| Leukemia & Lymphoma | 3.33 (2.37, 4.29) | 3.23 (1.60, 4.87) | 3.27 (1.71, 4.83) | 3.41 (2.01, 4.81) | |
| All others | 37.53 (35.21, 39.85) | 27.51 (24.03, 30.98) | 35.03 (30.79, 39.28) | 44.12 (40.00, 48.25) | |
| Education (%) | | | | | < .001 |
| <High school | 14.40 (12.30, 16.50) | 34.49 (29.80, 39.18) | 15.86 (12.48, 19.23) | 3.60 (2.30, 4.90) | |
| High school diploma or equivalent | 21.40 (18.98, 23.82) | 26.93 (23.01, 30.85) | 28.33 (23.72, 32.94) | 13.99 (10.86, 17.11) | |
| ≥College or above | 64.20 (60.99, 67.41) | 38.58 (33.24, 43.92) | 55.81 (51.21, 60.41) | 82.41 (79.1, 85.72) | |
| Marital Status (%) | | | | | < .001 |
| Married or living with partner | 67.21 (64.54, 69.88) | 46.16 (41.22, 51.11) | 64.99 (60.78, 69.21) | 79.00 (75.33, 82.66) | |
| Smoking status (%) | | | | | < .001 |
| Currently smoking | 17.09 (15.09, 19.09) | 28.90 (23.7, 34.09) | 17.23 (13.45, 21.00) | 11.23 (8.34, 14.12) | |
| Health insurance (%) | | | | | < .001 |
| Not-covered | 19.86 (16.96, 22.76) | 27.80 (22.46, 33.13) | 22.66 (18.12, 27.21) | 14.08 (11.26, 16.89) | |
| Alcohol consumption (%) | | | | | < .001 |
| ≥12 drinks/year | 73.57 (70.93, 76.21) | 62.40 (58.91, 65.9) | 68.90 (64.88, 72.92) | 82.20 (78.91, 85.50) | |
| Body mass index (%) | | | | | .061 |
| <25 kg/m$^2$ | 29.98 (27.84, 32.12) | 29.41 (26.07, 32.75) | 28.00 (24.24, 31.76) | 31.60 (27.43, 35.77) | |
| 25 - <30 kg/m$^2$ | 35.13 (32.43, 37.82) | 30.19 (26.28, 34.1) | 37.49 (32.24, 42.73) | 35.94 (31.47, 40.40) | |
| ≥30 kg/m$^2$ | 34.90 (32.75, 37.04) | 40.4 (36.33, 44.47) | 34.51 (30.38, 38.65) | 32.46 (29.21, 35.72) | |
| Walking difficulty score | 1.19 (1.17, 1.21) | 1.38 (1.33, 1.43) | 1.20 (1.16, 1.23) | 1.09 (1.06, 1.11) | < .001 |
| HEI 2015 | 53.11 (52.3, 53.91) | 49.82 (48.58, 51.06) | 52.97 (51.61, 54.34) | 54.76 (53.57, 55.96) | < .001 |
| MVPA levels (%) | | | | | < .001 |
| I-MVPA (<150 mins/wk) | 49.45 (47.18, 51.71) | 60.89 (56.74, 65.04) | 55.07 (50.99, 59.14) | 40.04 (36.33, 43.75) | |
| CVD Status (%) | | | | | < .001 |
| No-CVD | 83.2 (81.32, 85.09) | 76.01 (72.53, 79.5) | 78.89 (75.39, 82.39) | 89.65 (87.47, 91.83) | |

*(Continued)*

**Table 1.** (Continued)

| | Total | Poverty-income ratio[a] | | | P-value[b] |
|---|---|---|---|---|---|
| | | Lowest tertile | Middle tertile | Highest tertile | |
| Pre-existing CVD | 6.38 (5.39, 7.38) | 8.32 (6.07, 10.56) | 7.84 (5.91, 9.77) | 4.44 (3.02, 5.87) | |
| Post-acquired CVD | 10.42 (8.82, 12.01) | 15.67 (12.42, 18.92) | 13.27 (10.5, 16.04) | 5.91 (4.13, 7.68) | |

Abbreviation: CVD = cardiovascular disease; I-MVPA = insufficient moderate and vigorous-intensity physical activity (<150 minutes of MVPA per week);

HEI = healthy eating index

Values are the mean (95% CI) and the percentage (95% CI) for continuous and categorical variables, respectively, estimated after accounting for the complex sampling design of the NHANES.

[a]The thresholds of poverty-income ratio were 1.72 and 3.71.

[b] P-value is for the between-group differences estimated from a linear regression model for a continuous variable and Rao-Scott x2 test of independence for a categorical variable.

[c] Weighted population % indicates the prevalence estimates among US adults aged ≥18 years.

[d] 1st diagnosed cancer type, if multiple cancers were reported.

to participate in MVPA. Furthermore, participants in the lowest PIR tertile had a higher prevalence of obesity-related cancers, tobacco-related cancers, breast cancer, and colon cancer. As the PIR tertile increased, the prevalence of pre-existing and post-acquired CVD decreased. In the lowest PIR tertile, the prevalence of CVD was 8.3% for pre-existing and 15.7% for post-acquired disease, versus 4.4% and 5.9% in the highest PIR tertile. S1 Table in S1 File presents the descriptive characteristics of cancer survivors categorized by CVD status, with notable differences in sociodemographic characteristics by CVD status. Participants with post-acquired CVD had the lowest prevalence of college or above education, highest prevalence of body mass index equal to or over 30 kg/m$^2$, and lowest physical activity levels.

The cancer survivors in the lowest PIR tertile had the highest prevalence of pre-existing stroke (44.5% vs. 20–26% in the middle and highest tertile). Table 2 reports the prevalence of pre-existing and post-acquired CVD. The prevalence of post-acquired CVD, the number of years since first diagnosed with CVD, number having two or more cardiovascular conditions, heart failure, and stroke were higher in the lowest-tertile compared to the highest-tertile. However, these comparisons did not reach statistical significance.

We conducted a multivariate Cox regression analysis to assess the association between PIR and CVD with all-cause mortality in cancer survivors (Table 3). After adjusting for study covariates, including age, sex, race/ethnicity, cancer type, marital status, smoking, physical activity, body mass index, and walking difficulty score, that were retained using a backward elimination method with a criterion of $P < .20$, we observed that both CVD status and PIR tertile were independently associated with all-cause mortality. The results showed that, compared to cancer survivors without CVD, those with post-acquired CVD (HR = 1.67; 95% CI = 1.30–2.15) had a higher risk of all-cause mortality. Among PIR tertiles, cancer survivors in the lowest PIR tertile (HR = 1.71; 95% CI = 1.24–2.36) and middle PIR tertile (HR = 1.49; 95% CI = 1.04–2.14) had a significantly higher risk of all-cause mortality compared to those in the highest PIR tertile ($P$-for-trend $< .001$). When testing interaction term between PIR and CVD status, no statistically significant was found ($P>0.05$).

In the follow-up multivariate Cox regression analysis testing the combined association of PIR and CVD status (Table 4) demonstrated that, among cancer survivors with no CVD, we observed a 72% increase in risk in mortality between the lowest and highest PIR tertile (HR = 1.72; 95% CI = 1.69–4.35). Similarly, additional comparisons with alternative reference groups (S3 Table in S1 File) showed that, in cancer survivors with pre-existing CVD, the lowest

**Table 2. Prevalence and types of cardiovascular diseases among cancer survivors (aged≥18 years) By poverty-income ratio (NHANES 2003–2014).**

| | Total | Poverty-income ratio | | | P-value[a] |
| --- | --- | --- | --- | --- | --- |
| | | Lowest tertile | Middle tertile | Highest tertile | |
| Pre-existing CVD[b] | | | | | |
| 2+ CVDs (%) | 27.13 (18.75, 35.50) | 31.78 (19.76, 43.80) | 29.15 (13.44, 44.85) | 20.44 (5.10, 35.79) | .555 |
| CAD (%) | 69.42 (63.05, 75.78) | 62.60 (51.16, 74.05) | 74.85 (63.89, 85.81) | 69.13 (51.55, 86.72) | .302 |
| HF (%) | 32.93 (26.09, 39.77) | 33.36 (20.22, 46.50) | 32.13 (16.82, 47.43) | 33.49 (15.84, 51.14) | .989 |
| Stroke (%) | 29.73 (24.67, 34.79) | 44.51 (30.94, 58.07) | 26.18 (15.41, 36.94) | 20.47 (6.03, 34.91) | .033 |
| Post-acquired CVD[c] | | | | | |
| Years since first diagnosed with CVD after cancer[d] | 9.57 (8.26, 10.88) | 9.69 (7.66, 11.71) | 9.96 (7.79, 12.12) | 8.82 (6.83, 10.81) | .078 |
| 2+ CVDs (%) | 31.11 (24.59, 37.62) | 32.25 (21.22, 43.28) | 35.51 (25.20, 45.82) | 22.91 (13.30, 32.52) | .216 |
| CAD (%) | 54.01 (46.10, 61.92) | 57.26 (46.59, 67.93) | 46.33 (34.04, 58.63) | 61.51 (47.70, 75.32) | .094 |
| HF (%) | 32.98 (27.18, 38.78) | 32.69 (23.77, 41.62) | 36.19 (25.69, 46.70) | 28.43 (16.54, 40.32) | .500 |
| Stroke (%) | 36.56 (30.79, 42.33) | 40.89 (29.68, 52.10) | 40.00 (30.63, 49.37) | 25.69 (14.41, 36.97) | .086 |

Abbreviation: CAD = coronary artery disease; CVD = cardiovascular disease; HF = heart failure

Values are the percentage (95% CI) for all variables, with an exception of 'Years since diagnosed with CVD after cancer", which is presented using the mean (95% CI).

All values are estimated after accounting for the complex sampling design of the NHANES.

[a] P-value is for the between-group differences estimated from a linear regression model for a continuous variable and Rao-Scott x2 test of independence for a categorical variable.

[b] Cancer survivors who already had pre-existing cardiovascular diseases before diagnosed with cancer.

[c] Cancer survivors who acquired the cardiovascular diseases after diagnosed with any cancer.

[d] The time elapsed until they were first diagnosed with cardiovascular diseases after they were diagnosed with any cancer. When multiple CVDs were reported, the CVD diagnosed first was used to calculate average years.

and middle-PIR tertile groups continued to have a higher risk of mortality than the highest PIR tertile by 200% (HR = 3.00; 95% CI = 1.29–7.01) and 164% (HR = 2.64; 95% CI = 1.06, 6.61), respectively. These findings collectively suggest that PIR may be a potent predictor of mortality as pre-existing CVD and even a stronger predictor than post-acquired CVD. Lastly, the between-group difference in the risk of mortality by CVD status was apparent among

**Table 3. Associations of poverty-income ratio and cardiovascular disease status with all-cause mortality among cancer survivors.**

| | Unweighted deaths/total n | Weighted % of deaths (95% CI) [a] | Hazard ratio (95% CI) [b] |
| --- | --- | --- | --- |
| PIR | | | |
| Lowest tertile | 212/821 | 21.75 (18.02, 25.47) | 1.71 (1.24, 2.36) |
| Middle tertile | 207/815 | 19.01 (15.79, 22.24) | 1.49 (1.04, 2.14) |
| Highest tertile | 108/828 | 7.50 (5.47, 9.52) | Ref |
| | | | P-for-trend < .001[c] |
| CVD Status | | | |
| No-CVD | 317/1922 | 10.68 (9.17, 12.19) | Ref |
| Pre-existing CVD | 78/211 | 29.30 (20.96, 37.64) | 1.35 (1.00, 1.84) |
| Post-acquired CVD | 132/331 | 34.21 (27.78, 40.64) | 1.67 (1.30, 2.15) |

Abbreviation: CVD = cardiovascular disease; PIR = poverty-income ratio

[a] Weighted % of deaths represents the population-level prevalence (%) of all-cause deaths among cancer survivors.

[b] Hazard ratios are estimated from a Cox proportional hazard regression model, which includes both PIR and CVD status without an interaction term, while adjusting for age, sex, race/ethnicity, cancer type, marital status, smoking, physical activity, body mass index, and walking difficulty score that were retained by the backward elimination method (P < .20). The full multivariate model is presented in S2 Table in S1 File.

[c] The P-value for a linear trend of PIR tertiles was estimated using orthogonal polynomial contrasting.

**Table 4. Combined associations of poverty-income ratio and CVD status with all-cause mortality among cancer survivors.**

| PIR | CVD Status | Unweighted deaths/total *n* | Weighted % of deaths (95% CI) | Hazard Ratios (95% CI)[a,b] |
|---|---|---|---|---|
| Highest tertile | No-CVD | 72/710 | 6.10 (4.33, 7.87) | Ref |
| | Pre-existing CVD | 11/46 | 12.06 (3.07, 21.05) | 0.79 (0.38, 1.68) |
| | Post-acquired CVD | 25/72 | 25.26 (13.89, 36.63) | 1.54 (0.91, 2.62) |
| Middle tertile | No-CVD | 117/610 | 13.53 (10.45, 16.62) | 1.27 (0.88, 1.82) |
| | Pre-existing CVD | 33/80 | 37.89 (24.46, 51.32) | 2.10 (1.24, 3.54) |
| | Post-acquired CVD | 57/125 | 40.44 (30.98, 49.90) | 2.71 (1.69, 4.35) |
| Lowest tertile | No-CVD | 128/602 | 17.61 (13.84, 21.38) | 1.72 (1.21, 2.45) |
| | Pre-existing CVD | 34/85 | 36.90 (23.57, 50.23) | 2.38 (1.37, 4.14) |
| | Post-acquired CVD | 50/134 | 33.78 (21.93, 45.62) | 2.17 (1.27, 3.71) |

Abbreviation: CVD = cardiovascular disease; PIR = poverty-income ratio

[a] Hazard ratios are estimated from a Cox proportional hazard regression model, which includes a single combined variable (PIR-by-CVD status), while adjusting for age, sex, race/ethnicity, cancer type, marital status, smoking, physical activity, body mass index, and walking difficulty score that were retained by the backward elimination method (*P* < .20).

[b] Hazard ratios estimated with alternative reference categories are presented in S3 Table in S1 File. The full multivariate model is presented in S4 Table in S1 File.

cancer survivors in the middle PIR tertile, where the pre-existing (HR = 1.66; 95% CI = 1.05–2.61) and post-acquired (HR = 2.14; 95% CI = 1.49–3.07) CVD groups had significantly greater mortality risk when compared to the no-CVD group. However, such between-group differences by CVD status were not observed in the highest and lowest PIR tertile groups.

## Discussion

In this analysis of a large nationally representative sample of US adults between 2003 and 2014, we found that low income, as defined by PIR, was associated with an increased prevalence of CVD and increased risk of all-cause mortality among cancer survivors. Importantly, our study observed that low income may be as stronger predictor for mortality as CVD status in cancer survivors, emphasizing the role of income inequality as a driver of survival differences in cancer survivors, highlighting the need to consider socioeconomic factors in outcomes for cancer survivors.

To our knowledge, this is the first study to examine the impact of income on CVD risk and mortality across all individuals with a history of cancer. Prior studies have shown that low income is associated with poorer health behaviors, health status, and increased CVD risk in the general population. Recently, Minhas et al observed significant associations between PIR and adverse cardiovascular outcomes in NHANES participants [28]. In adjusted analysis, the prevalence of diabetes, hypertension, coronary artery disease, heart failure and stroke decreased in a stepwise manner from the highest to lower PIR [28]. In a separate NHANES analysis, lower PIR was associated with all-cause mortality independent of demographic, lifestyle, and clinical risk factors in the general population [29]. In our study of NHANES cancer survivors, these findings were re-demonstrated with the lowest PIR group reporting the highest prevalence pre-existing CVD and post-acquired CVD. Our analysis unveiled several possible mechanisms behind the highest prevalence of CVD among cancer survivors with lower PIR, such as the lower prevalence of healthy behaviors (higher percentages of obesity, tobacco use, walking difficulty score) that may contribute to worsening cardiovascular health [30–33]. Additional contributing factors include differences in insurance status, financial strain, and variations in access to prevention services [3, 34–37] by income status. Of note. our lowest PIR tertile had the highest prevalence of uninsured adults. Uninsured adults are less likely to

receive treatment for chronic conditions, are more likely to have limited access to care, and are more likely to have decreased utilization of recommended healthcare services [34, 38–40], which may be the result of income disparities [41]. The subsequent increase in cardiometabolic risk and increased vulnerability to chronic conditions may be accentuated in cancer survivors, emphasizing the need to improve risk mitigation in this population.

A novel finding in this present study is the impact of income on all-cause mortality in cancer survivors. We observed that PIR level was as strong a predictor of mortality as pre-existing CVD was in our highest PIR groups, and a stronger predictor of mortality than post-acquired CVD. This multifaceted phenomenon is driven by a complex interplay of factors, encompassing income, health behaviors, chronic conditions, discrimination, and healthcare access [25, 42–50]. Cancer survivors are an especially vulnerable population to CVD due to an increased prevalence of CVD risk factors, socioeconomic risk factors, and risks associated with cancer-related therapies [4, 13, 14]. It is essential to improve our understanding of how income and other socioeconomic factors affect CVD and mortality to improve health equity amongst vulnerable populations.

While our study provides novel insights, it is important to acknowledge several limitations. Our data is derived from a nationally representative sample of US adults, but the applicability outside of the US may be limited due to differences in patient populations, living conditions, and healthcare systems. Another important limitation is that PIR was not repeatedly measured over time; hence, we could not examine the time-dependent effects of PIR on the risk of mortality. Additionally, our cross-sectional data on CVD risk prevents us from establishing causal relationships. Self-reported behavior and medical history, including history of cancer, may entail misclassification and/or underreporting. Although NHANES is regarded as a valid assessment tool [11, 51], with several recent publications using self-reported cancer history [52, 53]. While we observed that lower PIR is associated with a higher prevalence CVD and increased risk of all-cause mortality, underlying mechanisms are complex, and we are unable to ascertain the exact pathways leading to this association. All-cause mortality in cancer survivors encompasses both cancer-related mortality and non-cancer related mortality. Income may differentially impact different cancer subtypes and mortality risks, with the majority of studies reporting that low household income is associated with worse outcomes [54, 55]. Unfortunately, due to the limited sample size, we were unable to broaden our scope to include cause-specific mortality, including CVD-specific mortality. However, several large studies have reported that CVD is the most common cause of non-cancer death, likely secondary to cardiotoxicity [56, 57], allowing us to highlight that income is an important risk measure for adverse health outcomes in this population.

In the future, it would be interesting to delve into patient level and clinical factors to further evaluate the connection between income, CVD risk, and cause-specific mortality in cancer survivors. More research is needed to deepen our understanding of the long-term interplay between income, CVD risk, and cancer outcomes in the United States.

## Conclusion

In conclusion, in this large cohort of participants with a history of cancer, we found that low income was associated with a higher risk of CVD and all-cause mortality. Income emerges as strong of a predictor of mortality, rivaling the influence of CVD in cancer survivors. Future research is needed to better understand the impact of socioeconomic risk factors with outcomes in this population. Public health interventions and policies focused on cardiovascular disease and cancer survivors must take income disparities into account.

## Supporting information

**S1 File. Supplement tables.**
(DOCX)

## Author Contributions

**Conceptualization:** Vidhushei Yogeswaran, Youngdeok Kim, Richard K. Cheng.

**Data curation:** Vidhushei Yogeswaran, Youngdeok Kim.

**Formal analysis:** Vidhushei Yogeswaran, Youngdeok Kim, Richard K. Cheng.

**Investigation:** Vidhushei Yogeswaran, Youngdeok Kim, R. Lee Franco, Alexander R. Lucas, Arnethea L. Sutton, Jessica G. LaRose, Jonathan Kenyon, Ralph B. D'Agostino, Jr., Vanessa B. Sheppard, Kerryn Reding, W. Gregory Hundley, Richard K. Cheng.

**Methodology:** Vidhushei Yogeswaran, Youngdeok Kim, Richard K. Cheng.

**Validation:** Richard K. Cheng.

**Writing – original draft:** Vidhushei Yogeswaran, Youngdeok Kim.

**Writing – review & editing:** Vidhushei Yogeswaran, Youngdeok Kim, R. Lee Franco, Alexander R. Lucas, Arnethea L. Sutton, Jessica G. LaRose, Jonathan Kenyon, Ralph B. D'Agostino, Jr., Vanessa B. Sheppard, Kerryn Reding, W. Gregory Hundley, Richard K. Cheng.

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
