## [Decision Letter · Decision Letter 0]

16 Nov 2023

PONE-D-23-28914Association of Income Level with Cardiovascular Disease and Mortality in Cancer Survivors in the United StatesPLOS ONE

Dear Dr. Kim,

Thank you for submitting your manuscript to PLOS ONE. After careful consideration, we feel that it has merit but does not fully meet PLOS ONE’s publication criteria as it currently stands. Therefore, we invite you to submit a revised version of the manuscript that addresses the points raised during the review process.

Several points by our reviewers require further clarification in order to be understood by readers. In particular, your hypothesesis should be clarified and how these data contributed to answer this question. Please also clarify why no CVD-related deaths could not be addressed in your analysis. Some minor points (use of "cohort" in a cross-sectional analysis, as well as the construcztion of your statistical model (inclusion of confounders) should be re-checked thoroughly. 

We look forward to receiving your revised manuscript.

Kind regards,

Thomas Behrens

Academic Editor

PLOS ONE

Reviewers' comments:

Reviewer's Responses to Questions

**Comments to the Author**

1. Is the manuscript technically sound, and do the data support the conclusions?

Reviewer #1: Partly

Reviewer #2: Partly

Reviewer #3: Partly

2. Has the statistical analysis been performed appropriately and rigorously? 

Reviewer #1: I Don't Know

Reviewer #2: Yes

Reviewer #3: I Don't Know

3. Have the authors made all data underlying the findings in their manuscript fully available?

Reviewer #1: Yes

Reviewer #2: Yes

Reviewer #3: Yes

4. Is the manuscript presented in an intelligible fashion and written in standard English?

Reviewer #1: No

Reviewer #2: Yes

Reviewer #3: No

5. Review Comments to the Author

Reviewer #1: Yogeswaran et al. have analysed the interesting topic on the association of income, cancer, cardiovascular disease (CVD), and mortality. However, in this manuscript the relationship between introduction, the methods used, and interpretation of the results is not straightforward in my view. Here are the details, along with some other comments:

1. The analysis focuses on cancer survivors, as this group is known to have elevated CVD risks. These risks have been attributed to usual CVD risk factors and (adverse) effects of cancer therapy, so far, and become “a critical cause for morbidity and mortality”. (Which further morbidity, is it relevant here?) Although this points to CVD-related mortality, the endpoint defined in this analysis is all-cause mortality. Was no information available on causes of mortality? All-cause mortality introduces more complexity, and income is a risk factor for many causes of death. Thus, one would expect differences in all-cause mortality by income even when controlling for CVD. The authors do not sufficiently introduce or discuss these pathways (which covariates are supposed to be confounders, which ones mediators?), and do not discuss different causes of mortality at all. Accordingly, the regression models are based on statistical significance rather than on knowledge about the pathways.

2. The discussion lacks comparison of the results with other studies, and especially with the NHANES-population without cancer. Are the differences by income in the ‘general’ NHANES population the same for ‘No-CVD’ and participants with CVD diagnosed before cancer?

3. In part, I could not completely understand how regression analysis was conducted, or to which results it refers: In the methods section, 2 models are introduced, the first includes (among others) both, income and CVD. But table 3 does not list the respective other variable? In my opinion, the mutual adjustment would be essential here. The second model should additionally include an interaction term (‘joint effect’), or does it refer to a further stratified variable as shown in table 4?

4. Maybe related to this: I cannot follow the conclusion that income is a stronger predictor than CVD. Which results do support this?

5. The terminology/language in the manuscript are partially not consistent and precise enough. E.g., income is measured on individual, household or area level, and there are absolute or relative measures, all of which may be differently associated with health outcomes. The manuscript does not seem to differentiate in this respect. Another example is use of terms like ‘independently associated’ all over the manuscript: it is often not clear, what exactly this means.

6. In the classification of cancer types, there are some overlaps. How did the authors decide to classify e.g. lung cancer? What was the procedure in case of more than one cancer diagnosis (also with respect to follow-up time?). This should be mentioned in the methods section.

7. Overall, income is differently associated with different cancer types. The analysis does not consider this (probably due to small numbers), but it should at least be discussed.

8. Sex/gender is associated in every (biological, social/behavioural) respect. Therefore, I would suggest a stratified analysis by sex (at least supplementary).

9. Is there a reference for the cut point (p=0.2) for backward elimination? It seems rather high, and possibly a reason why nearly all initial covariates are included.

10. For mortality, follow-up years are only mentioned in methods, what was the starting point?

11. Tables/figures:

- Table 1 should additionally include data on mortality, NHANES-waves, and follow-up years/person-years at risk.

- In tables 3 and 4, there are columns with alternative reference groups and hyphens for some categories. Are results just omitted for some purpose?

- Figure 2 could be omitted, if it shows the same results as table 3.

12. The presentation of results of table 2 in the text seems to be wrong and misleading: There is no general elevated prevalence in the lowest income tertile. I don’t see any common pattern overall.

13. Numbers for middle income tertile are different in the text as compared to corresponding table (p 9 lines 19 and 21)

14. On page 12, authors note the ‘nationally representative’ US sample, and then state a limitation is that the ‘demographics in the actual cohort are representative of the US population’. This seems to be contradictory.

Reviewer #2: In the work presented, the aim was to determine the relationship between CVD and income inequality among cancer survivors. In my view, the motivation for this project is not clearly stated in the introduction. Why is this correlation particularly interesting among cancer survivors? Wouldn't it be more interesting to compare this influence in people with and without cancer to understand whether it differs between the groups? Without taking into account people without cancer, I also think the statement that the increased cardiometabolic risk and vulnerability may be exacerbated in cancer survivors is not necessarily valid.

Furthermore, the presentation in the tables with the results of the statistical models is not clear. I cannot understand what exactly was calculated, which covariates are included in the models and how the calculated models differ. As the full model results are not shown, it is not possible to assess the effects of the extensive adjustment variables.

Moreover, I cannot always understand the conclusions drawn from the results (e.g., that low income is a stronger predictor of mortality than CVS). Looking at Figure 2, the two risk factors appear to me to be similarly strong.

Below are more detailed comments:

Abstract: The abstract should include more details on the methods used. At this point, the previously unexplained abbreviation "HR" could also be introduced.

Methods:

The inclusion and exclusion criteria could be presented in more detail (page 5). For example, was there an inclusion criterion regarding a minimum or maximum length of time since cancer diagnosis? Perhaps people with childhood cancer diagnoses have a different CVD risk than other cancer survivors? This should be discussed.

Please also specify “valid responses to questions regarding the types of cancer and ages” (page 5, line 3).

Page 6: Please provide at least rudimentary information on the healthy eating index (e.g. range, are higher or worse values good, etc.).

Page 7:

I do not understand the following sentence: „We particularly estimated the prevalence of cancer survivors with 2 or more CVDs and type of CVD (i.e., CAD, heart failure, and stroke) across the time when CVD was diagnosed (i.e., pre-existing before cancer diagnosis or post-acquired CVD after cancer diagnosis).”

How are multiple CVDs measured? I had understood the previous paragraph to mean that people are always assigned to one of three CVD groups. Please clarify.

Results:

Page 8:

• I stumble over the expression "a final analytical sample". It sounds like there are several. Is this the case?

• Are there covariates with p>.20 included in the final model after variable selection? (line 6). Please clarify. Also, it is not clear from the results presented which covariates were included in the model.

Page 9:

• Presenting the estimators of the complete model shown in Figure 2 in the appendix would give the reader the opportunity to look at the magnitudes of the estimators of the influencing factors and the confounders.

• Why are pre-existing CVD and post-acquired CVD independently associated with mortality? Please clarify "independently".

• In contrast to the presentation in Table 3, why is the lowest PIR mentioned as a reference in the text? This should be consistent.

Table 3: Why are there two columns of hazard ratio? Are the HRs in the right-hand column based on models in a subpopulation, or were the PIRs combined for the middle and highest groups?

Page 10:

• I cannot evaluate the results of the interaction analysis based on the data presented.

• The statement that in every PIR tertile CVD increases the risk of mortality cannot be said with certainty with the presentation in Table 4. This is because the reference in the first model (column 5 of the table) is "No CVD in the highest PIR tertile" and applies to all combined PIR-CVD states. A stratified analysis might be more informative.

Table 4. As with Table 3, it is unclear to me why different models were presented and on which study population they are based.

Discussion:

Where are the results for the statement that low income was independently associated with an increased risk of CVD? I have only seen models with all-cause mortality as the outcome variable. Table 2, on the other hand, does not show adjusted risk estimates but prevalences.

I also cannot understand the statement that low income is a stronger predictor of mortality than CVS when I look at Table 3 (page 13, lines 14-15).

Page 11:

Due to the lack of presentation of the influences of the covariates, the discussion cannot always be compared with the study results (e.g., with regard to the status of health insurance). The way it is presented in the discussion, and I can also imagine it, the lack of health insurance has an essential influence on general health. This aspect is discussed, but not substantiated with figures from the study in relation to the endpoint of all-cause mortality.

Please specify “higher income groups”.

I have noticed several typos and missing spaces. Here are a few:

Page 4, line 2: “.(15).”

Page 5, line 23: Shouldn’t it be “college or above” rather than “≥college or above”?

Page 10, line 13: “that that”

Page 11, line 5: “death(3).“

Page 18: Figure legends should be given for Figure 1 and Figure 2.

Reviewer #3: Relevant and clearly structured manuscript. The topic is well introduced. However, some major and minor shortcomings should be addressed.

1. At the end of the introduction (p4, l. 2-12), a hypothesis is first formulated and then the questions and objectives of the study are stated. This passage of the manuscript is not clearly formulated and should be revised. The research question, hypothesis and aimes do not match. Specifically, the sentence on page 4, line 2 should be in more detail elaborated (e.g. reference to the limited information available, or reformulate the sentence in case no previous research has been done on the topic).

2. The study population is introduced as a repeated cross-sectional survey. In the section on the measurement of outcomes, it becomes clear that these observations contain retrospective information on CVD and are matched with mortality data in the future. This should be explained in the section on the study population and the follow-up time should be specified.

3. Figure 1 shows that more than 20% of the observations are not used due to missing observations. Are these missings systematic or random over the covariates?

4. Could you please explain the procedure you used to calculate a 12 year sample weight? (p.5, l.8,9) For which analysis did you use these sampling weights?

5. The concept of PIR (Poverty-Income ratio) is introduced to measure income. Please refer to the literature on this concept and please justify why you use this measure for categorization. Specifically, some arguments regarding classification into 2 (e.g. in poverty or not in poverty) or more groups, and arguments regarding alternative measures (e.g. adjusted household income) would be desirable.

6. Please describe your categories of the CVD measurement. What does 2+CVD mean?

7. Please describe your weighting procedure in the statistical analysis section. Did you use weighting only for descriptive statistics or also for the regression analysis?

8. Please refer to the literature on the backward elimination procedure. Specifically, how did you choose the 0.2 cut-off value? Is there a literature supporting this choice? Do you mean on page 8, line 6: P-value<.20? (Caution: typo)

9. Please explain your number of weighted cancer survivors (p8, l14). How can this number be interpreted?

10. Please refer in the results section always to a table or figure (p8. l14-20)

11. Isn't backward elimination a way of reducing covariates in the model rather than part of the regression analysis? (p.9. l.15)

12. You use several cohorts (P.9, l.17)

13. Please explain the findings of Table 3 before you refer to them.

14. Tables 3 and 4 and Figure 2 are not self-explanatory. How can "weighted % of deaths" be interpreted? Are the HRs of PIR and CVD status (Table 3, Figure 2) estimated simultaneously in one model or in separate models? What is the meaning of the different models in Tables 3 and 4? Please explain the meaning of independent and combined in this context. Please explain which hypothesis you are testing in which table/model. Overall, the problems in (1) above continue here. This part of the manuscript is also unclear and should be revised.

15. The text and the numbers related to Table 3 do not match (page 9, lines 19-22). Please, correct them.

In sum I come to the conclusion, that the paper is unsuitable for publication in its present form. However, the study itself shows sufficient potential for publication. Therefore, the authors are encouraged to resubmit a revised version.

6. PLOS authors have the option to publish the peer review history of their article (what does this mean?). If published, this will include your full peer review and any attached files.

Reviewer #1: No

Reviewer #2: No

Reviewer #3: No

---

## [Author Response · Author response to Decision Letter 0]

18 Dec 2023

Thank you so much for taking the time to read our paper and to give us feedback on this important topic. We have responded to the individual comments line-by-line in the attachment and hope the manuscript is now acceptable for publication.

---

## [Decision Letter · Decision Letter 1]

8 Jan 2024

PONE-D-23-28914R1Association of Poverty-income Ratio with Cardiovascular Disease and Mortality in Cancer Survivors in the United StatesPLOS ONE

Dear Dr. Kim,

Thank you for submitting your manuscript to PLOS ONE. After careful consideration, we feel that it has merit but does not fully meet PLOS ONE’s publication criteria as it currently stands. Therefore, we invite you to submit a revised version of the manuscript that addresses the points raised during the review process.

Please elaborate  more about  the covariates and also possible collinearity

We look forward to receiving your revised manuscript.

Kind regards,

Hamid Reza Baradaran, M.D., Ph.D.,

Academic Editor

PLOS ONE

Additional Editor Comments:

Please elaborate more about the covariates and also possible collinearity

Reviewers' comments:

Reviewer's Responses to Questions

**Comments to the Author**

1. If the authors have adequately addressed your comments raised in a previous round of review and you feel that this manuscript is now acceptable for publication, you may indicate that here to bypass the “Comments to the Author” section, enter your conflict of interest statement in the “Confidential to Editor” section, and submit your "Accept" recommendation.

Reviewer #4: All comments have been addressed

Reviewer #5: (No Response)

2. Is the manuscript technically sound, and do the data support the conclusions?

Reviewer #4: Yes

Reviewer #5: Partly

3. Has the statistical analysis been performed appropriately and rigorously? 

Reviewer #4: Yes

Reviewer #5: Yes

4. Have the authors made all data underlying the findings in their manuscript fully available?

Reviewer #4: Yes

Reviewer #5: No

5. Is the manuscript presented in an intelligible fashion and written in standard English?

Reviewer #4: Yes

Reviewer #5: Yes

6. Review Comments to the Author

Reviewer #4: (No Response)

Reviewer #5: Title: Ok

Abstract: I think it is necessary to define the study participants and the related used data base in more detail. This will affect the judgement of the readers about the possible bias and confounders.

Keywords: OK

Introduction: OK

Material & method: there are a number of issues to be noticed in this section:

- I think it is essential to show the reliability of the question used for the diagnosis of cancer in recruited patients. Another issue is the matter of the time of cancer diagnosis which shows the time to the event (outcome) of the study.

- Please define the inclusion and exclusion criteria in more detail. It seems mor criteria have been used for recruiting the participants.

- Please show how the parameters used for the calculation of PIR have been measured.

- The matter of measuring method of the covariates remains an important issue in the measurement of covariates. Another issue to be noticed is the probable collinearity of these variables with exposure of interest (PIR), which may cause problems for the analysis of the data.

- Showing the reliability and validity of measuring the outcomes of interest, is essential.

Results and discussion: according to the above-mentioned comments these sections are not justifiable.

7. PLOS authors have the option to publish the peer review history of their article (what does this mean?). If published, this will include your full peer review and any attached files.

Reviewer #4: **Yes: **Elham Mohebbi

Reviewer #5: **Yes: **Babak Eshrati

---

## [Author Response · Author response to Decision Letter 1]

17 Jan 2024

The respond to reviewers comments are attached to this submission.

---

## [Decision Letter · Decision Letter 2]

5 Feb 2024

PONE-D-23-28914R2Association of Poverty-income Ratio with Cardiovascular Disease and Mortality in Cancer Survivors in the United StatesPLOS ONE

Dear Dr. Kim,

Thank you for submitting your manuscript to PLOS ONE. After careful consideration, we feel that it has merit but does not fully meet PLOS ONE’s publication criteria as it currently stands. Therefore, we invite you to submit a revised version of the manuscript that addresses the points raised during the review process.

There are some methodological issues which have been remained  

We look forward to receiving your revised manuscript.

Kind regards,

Hamid Reza Baradaran, M.D., Ph.D.,

Academic Editor

PLOS ONE

Journal Requirements:

Additional Editor Comments:

There are some methodological issues which have been remained

Reviewers' comments:

Reviewer's Responses to Questions

**Comments to the Author**

1. If the authors have adequately addressed your comments raised in a previous round of review and you feel that this manuscript is now acceptable for publication, you may indicate that here to bypass the “Comments to the Author” section, enter your conflict of interest statement in the “Confidential to Editor” section, and submit your "Accept" recommendation.

Reviewer #5: All comments have been addressed

2. Is the manuscript technically sound, and do the data support the conclusions?

Reviewer #5: Partly

3. Has the statistical analysis been performed appropriately and rigorously? 

Reviewer #5: No

4. Have the authors made all data underlying the findings in their manuscript fully available?

Reviewer #5: Yes

5. Is the manuscript presented in an intelligible fashion and written in standard English?

Reviewer #5: Yes

6. Review Comments to the Author

Reviewer #5: Thanks to the revisions performed by the distinguished authors, I think there are still a number of issues, necessary to be noticed in this article:

- It is very important to show the reliability of measuring a number of variables including measurement of economic status of the recruited participants which are based on their response.

- According to what is mentioned in the material and method section, the study is a historical cohort one, in that case expression of the results shown in the first paragraph of the result section is meaningless. The two study groups are those with higher and lower PIR tertile not the CVD ones.

- Because of the nature of the study, it is necessary to show how probable selection biases such as immortal time bias, have been managed in this study. In fact, the differences shown in two groups of PIR status might have been related to the general condition differences of these two groups not to the proposed variable of PIR. This is shown somehow in table 1. In this case the conclusions may need to be revised totally.

- Another issue to be noticed in this study is the fact that measurement of the PIR has been conducted after the diagnosis of both cancer and CVD, which may not be a correct reflection of the PIR status of the start of the study.

7. PLOS authors have the option to publish the peer review history of their article (what does this mean?). If published, this will include your full peer review and any attached files.

Reviewer #5: **Yes: **Babak Eshrati

---

## [Author Response · Author response to Decision Letter 2]

5 Feb 2024

The response letter to the reviewer's comments is attached in this submission.

---

## [Decision Letter · Decision Letter 3]

22 Feb 2024

Association of Poverty-income Ratio with Cardiovascular Disease and Mortality in Cancer Survivors in the United States

PONE-D-23-28914R3

Dear Dr. Kim,

We’re pleased to inform you that your manuscript has been judged scientifically suitable for publication and will be formally accepted for publication once it meets all outstanding technical requirements.

Kind regards,

Hamid Reza Baradaran, M.D., Ph.D.,

Academic Editor

PLOS ONE

Additional Editor Comments (optional):

Reviewers' comments:

Reviewer's Responses to Questions

**Comments to the Author**

1. If the authors have adequately addressed your comments raised in a previous round of review and you feel that this manuscript is now acceptable for publication, you may indicate that here to bypass the “Comments to the Author” section, enter your conflict of interest statement in the “Confidential to Editor” section, and submit your "Accept" recommendation.

Reviewer #5: (No Response)

2. Is the manuscript technically sound, and do the data support the conclusions?

Reviewer #5: Yes

3. Has the statistical analysis been performed appropriately and rigorously? 

Reviewer #5: Yes

4. Have the authors made all data underlying the findings in their manuscript fully available?

Reviewer #5: Yes

5. Is the manuscript presented in an intelligible fashion and written in standard English?

Reviewer #5: Yes

6. Review Comments to the Author

Reviewer #5: all comments have been addressed. the paper is ready to be published

7. PLOS authors have the option to publish the peer review history of their article (what does this mean?). If published, this will include your full peer review and any attached files.

Reviewer #5: **Yes: **Babak Eshrati

---

## [Editor Report · Acceptance letter]

26 Jun 2024

PONE-D-23-28914R3 

PLOS ONE

Dear Dr. Kim, 

I'm pleased to inform you that your manuscript has been deemed suitable for publication in PLOS ONE. Congratulations! Your manuscript is now being handed over to our production team.

Kind regards, 

on behalf of

Professor Hamid Reza Baradaran 

Academic Editor

PLOS ONE